# Strategies to Connect Low-Income Communities with the Proposed Sewerage Network of the Dhaka Sanitation Improvement Project, Bangladesh: A Qualitative Assessment of the Perspectives of Stakeholders

**DOI:** 10.3390/ijerph17197201

**Published:** 2020-10-01

**Authors:** Mahbub-Ul Alam, Fazle Sharior, Sharika Ferdous, Atik Ahsan, Tanvir Ahmed, Ayesha Afrin, Supta Sarker, Farhana Akand, Rownak Jahan Archie, Kamrul Hasan, Rosie Renouf, Sam Drabble, Guy Norman, Mahbubur Rahman, James B. Tidwell

**Affiliations:** 1Environmental Interventions Unit, Infectious Disease Division, icddr,b, Dhaka 1212, Bangladesh; fazle.sharior@icddrb.org (F.S.); sharika.ferdous@icddrb.org (S.F.); atikahsan@gmail.com (A.A.); ayesha.afrin@icddrb.org (A.A.); supta@icddrb.org (S.S.); dhulikona@gmail.com (F.A.); mahbubr@icddrb.org (M.R.); 2Department of Civil Engineering, Bangladesh University of Engineering and Technology, Dhaka 1000, Bangladesh; tanvirahmed@ce.buet.ac.bd; 3Institute of Statistical Research and Training, University of Dhaka, Dhaka 1000, Bangladesh; rownak.jahan.archie.13@gmail.com; 4Dhaka Water Supply and Sewerage Authority (DWASA), Dhaka 1215, Bangladesh; kamrul410s@gmail.com; 5Water & Sanitation for the Urban Poor (WSUP), London ECV4 6AL, UK; RRenouf@wsup.com (R.R.); SDrabble@wsup.com (S.D.); gnorman@wsup.com (G.N.); 6Harvard Kennedy School of Government, Cambridge, MA 02138, USA; bentidwell@gmail.com; 7World Vision Inc., Washington, DC 20002, USA

**Keywords:** urban sanitation, sewerage network, sewerage connection, low-income community, slum, DSIP, affordability, Dhaka, Bangladesh

## Abstract

In Bangladesh, approximately 31% of urban residents are living without safely managed sanitation, the majority of whom are slum residents. To improve the situation, Dhaka Water Supply and Sewerage Authority (DWASA) is implementing the Dhaka Sanitation Improvement Project (DSIP), mostly funded by the World Bank. This study assessed the challenges and opportunities of bringing low-income communities (LICs) under a sewerage connection within the proposed sewerage network plan by 2025. We conducted nine key-informant interviews from DWASA and City Corporation, and 23 focus-group discussions with landlords, tenants, and Community Based Organisations (CBOs) from 16 LICs near the proposed catchment area. To achieve connections, LICs would require improved toilet infrastructures and have to be connected to main roads. Construction of large communal septic tanks is also required where individual toilet connections are difficult. To encourage connection in LICs, income-based or area-based subsidies were recommended. For financing maintenance, respondents suggested monthly fee collection for management of the infrastructure by dividing bills equally among sharing households, or by users per household. Participants also suggested the government’s cooperation with development-partners/NGOs to ensure sewerage connection construction, operation, and maintenance and prerequisite policy changes such as assuring land tenure.

## 1. Introduction

### 1.1. Urban Sanitation and Consequences

Worldwide, 2 billion people still lack a basic sanitation service, a burden disproportionately borne by the world’s poor [1]. In urban areas globally, 47% of the population use safely managed sanitation services, 38% use basic services, and 9% use limited services [1]. According to the WHO-UNICEF Joint Monitoring Program (JMP) [2], improved sanitation facilities are those designed to hygienically separate excreta from human contact; improved sanitation facilities which are not shared with other households and the excreta produced are treated properly are considered safely managed sanitation services. If the excreta from improved sanitation facilities are not safely managed, then those facilities are classified as basic sanitation services; improved facilities, which are shared with other households, are classified as limited services. In low-income communities in Dhaka, only 2% of the population have access to safely managed sanitation [3], and no faecal material is considered safely managed outside of a small fraction going into the sewer network [4]. The Bangladeshi low- and middle-income communities usually earn between $2 to $20 per capita per day [5]. In low-income urban areas with poorly developed infrastructure, high population growth coupled with low socio-economic status leaves shared sanitation facilities often as the only viable sanitation option [6]. A common practice in low- and middle-income countries like Bangladesh is to connect flush/pour-flush toilets directly to drains, without any form of on-site containment [7]. Inadequate sanitation leads to environmental pollution and has significant adverse health [3,8,9] and non-health consequences [10]. The Sustainable Development Goals (SDG) 6 aims at ensuring the availability and sustainable management of water and sanitation for all. It is estimated that just the capital expenditures needed to extend water and sanitation services globally to meet the SDGs are $1.7 trillion up until 2030. Urban sanitation makes up 44% of these costs [11]. Generally, these costs are funded both from the government, along with other international agencies such as different development banks. Hereby, development banks’ investments in sanitation are usually not pro-poor and are failing to promote the transformational change that is needed to overcome the urban sanitation crisis [12].

### 1.2. Sewerage Facilities in Dhaka: Current State

In urban Dhaka, the existing sewerage network serves only 20% of the total urban population, mostly concentrated in wealthier areas. A 2005 census identified 4966 slums in Dhaka city and found that almost all were without any sewerage network coverage. Dhaka’s current population density is the 6th highest in the world—29,069 people per km^2^ (2020) [13]. Over 35% of the city’s population (18 million) lives in Dhaka’s low-income settlements. Improved sanitation facilities are mostly found among middle and higher-income households [14]. A rapid increase in urbanisation with insufficient Faecal Sludge Management (FSM) services and inadequate investment in the sector are major barriers to pro-poor urban sanitation [7]. Additionally, low-income households usually refrain from investments in sanitation due to tenure insecurity [15]. Many residents in low-income settlements use simple pit latrines with/without water seals, septic tanks, cluster latrines, communal latrines, or hanging sanitation suspended over water bodies [14]. Faecal matter from pits and septic tanks spills over into open areas, and stormwater in over-flowing open drains becomes contaminated with untreated sewage. This poses significant health risks to the poor, who often live in low lying or unplanned settlements [16]. The Bangladesh Bureau of Statistics (BBS) identifies urban LICs to predominantly be characterised by poor housing, poor quality or no sewerage and drainage, inadequate drinking water supply, and few or no paved streets or paths [17,18]. Many LICs are also located near polluted water bodies, swamps, or putrid drainage canals [19]. The urban poor population in this country is being neglected in many ways in terms of social, economic, and infrastructural improvement [20,21,22,23]. Limited access to water and sanitation services in slums could contribute to the poor health of slum residents [24].

### 1.3. DSIP and LICs: Unaddressed Issues

Dhaka Water Supply and Sewerage Authority (DWASA) is a Bangladesh government agency under the Ministry of Local Government, Rural Development and Co-operatives responsible for water and sewage in Dhaka. DWASA has planned to implement the Sewerage Master Plan under the Dhaka Sanitation Improvement Project (DSIP) with a loan of an estimated US$900 million from the World Bank [25]. As part of the master plan, by 2035, all households and public/private facilities are planned to have either access to the public sewerage system or improved on-site (or hybrid) facilities. The core objective of the Master Plan is wastewater management and improving sanitation systems in Dhaka City. However, this master plan does not clarify how connections to the sewerage network will be made for illegal settlements. A clearer illustration of financial support for those not being able to afford toilet renovation was also absent in the plan. Due to the per capita costs and estimated revenue that can be collected, households in the urban centres are proposed to be given the highest priority for sewerage access, whilst those in the transitional area will be served in stages. The remaining households situated in the on-site treatment area will remain un-sewered until after the target year of the master plan. Consequently, these households will be required to install individual on-site containment facilities or to develop a cluster-wise community sewerage system for combined treatment of night soil/septic tank sludge and sullage. 

Phase 2 of the Master Plan, spanning 2015–2025, aims to achieve that wastewater be collected from approximately 3.3 million people in the Pagla catchment and 1.4 million people in the Dasherkandi catchment, assuming an estimated household connection rate of 65% [26]. However, the master plan lacks clarification of an effective procedure to connect LICs in Dhaka city with the proposed sewerage network. Moreover, in urban Bangladesh, faecal sludge management services are not generally delivered through formal utilities but rather through well-functioning informal markets that are operated by middlemen and local providers for emptying on-site septic tanks [19]. In order to provide legal water connections to several areas in Dhaka, including LICs, regardless of tenure/holding numbers, DWASA has previously altered and updated its citizen charter. For such cases, an LIC department was also established, although informal markets continue to play a crucial role. Since legal service connections have been extended towards LICs in the past with resident’s contributions via incentives directed for cost-recovery and revenue collection for DWASA, such strategies could be adopted in the case of providing legal sanitation services to the low-income residents as well [27,28,29,30]. Recent data suggest that 42% of urban residents rely on on-site sanitation facilities [31]. However, Dhaka’s only treatment plant functions below capacity; due to several blockages and inefficiency issues of the lifting pumps, the treatment plant is not getting enough faecal waste to treat according to its full capacity. The existing network transports only 2% of the sewage produced, and only 0.3% is effectively treated [31]. According to the plans stated in the DWASA Sewerage Master Plan of Dhaka City for the period 2011–2035, sewer connections will be provided to all residents in Dhaka City as part of the DSIP. However, there are many challenges to achieving this goal, including cost, engineering design and construction, and modalities for financing and carrying out maintenance.

### 1.4. Study Objectives

This study aimed to assess the challenges and opportunities of connecting LICs to a sewerage network in the Pagla catchment area under DSIP. Many studies have been conducted regarding water supply and its pricing among slums in Dhaka [32,33,34] but less has been done for sanitation, especially focused upon users’ priorities to connect to a sewerage network in urban Dhaka. Notably, we are not aware of any research focusing on the priorities of toilet-users in low-income urban settlements to understand their preferences or strategies to connect them with the sewerage network. This paper presents the findings of the qualitative research regarding sewerage connections in low-income communities under the Dhaka Sanitation Improvement Project.

Here, we (a) ascertained the perceived benefits of the proposed sewerage network, (b) assessed the challenges and opportunities of connecting LICs to a sewerage network, and financing and maintaining those networks, (c) explored the perception of LIC residents about the affordability of connecting with the proposed sewerage network, and (d) explored the perceived barriers to ensuring a well-functioning sewerage system.

## 2. Methodology

### 2.1. Study Site

DWASA has planned a sewerage trunk main for the DSIP project to be implemented within the sewerage master plan. DWASA split this into two distinct parts: The Western trunk main and Eastern trunk main. To date, the project has mainly focused on the Eastern trunk main. Areas planned to cover through the current phase are located in Dhaka South City Corporation (DSCC); therefore, most of the study LICs were selected from DSCC. Additionally, we also chose two LICs from Dhaka North City Corporation (DNCC) to illustrate a comparison with the LICs located in DSCC. Hence, we selected study locations that were located within a distance of 2–3 km from the Eastern trunk main. The study carefully considered the distance of the settlements from the proposed trunk main since lack of detailed information regarding user perspectives and way-out to connect, nearby LICs should not be skipped from having sewerage connection. The study was conducted from February 2019 to February 2020. LICs were selected purposively by considering the following:Stratified by the size of the population: 11 LICs were selected where the number of households was less than 500, and 5 other LICs were chosen with more than 1000 households;Distance of the settlement from the proposed Eastern trunk main: 5 LICs within 500 m, 8 LICs > 500 m < 5 km, and 3 LICs > 5 km.

We selected a total of 16 LICs following these criteria from the slum list of Bangladesh Census of Slum Areas and Floating Population 2014 (Figure 1). The names of the selected areas were: Agargaon, Bhashantek, Dholpur, Duaripara, IG Gate Bank Colony, Kamlapur Railway, Kamrangirchar, Khilgaon Bagicha, Lalchan Mukim Lane, Maniknagar Adorsho Staff Quarter, Mogbazar Railway, Mohajer Colony, Nobinbag, Pagla, Shyampur, and Tekpara. Among these 16 LICs, 7 were situated on government owned land, 8 were on private land or on land that was leased by a businessman or political person, and the rest was residing on disputed land. Not all of the 7 LICs on government owned land had legal permission to extend their settlements. “IG Gate Bank Colony” and “Maniknagar Adorsho Staff quarter” areas had partial recognition to a certain level.

### 2.2. Study Design

We chose qualitative research approaches so that the study can capture the attitudes of the respondents. One of the core aims of the research was to illustrate user preferences. In this context, qualitative research techniques seemed best suited, as they are not bound by the methodological limitations and explain something which numbers merely are unable to reveal. These approaches allowed us to be far more speculative about what we have chosen to investigate. This qualitative exploratory study used Key Informant Interviews (KII) and Focus Group Discussions (FGD) for data collection. Collected data were analysed to identify the LICs respondents’ point of view about their needs and perceived benefits in terms of sewerage facilities, which were broadly unaddressed in previous studies. This study tried to represent the perspectives of LIC residents and a pathway for change to improve their lives.

### 2.3. Study Population

The study included two groups: authorities responsible for service delivery and residents in the LICs. We performed key informant interviews (KIIs) with authorities and focus group discussions (FGDs) with the three types of groups from LICs consisting of landlords, tenants, and Community Based Organization (CBO) leaders. FGD participants resided either on governmental land as an unregistered settlement or on land that is leased from the government by a third party. In the study, the group “landlord” means he/she owned the house only, or they were allocated to use the land for a certain time, or who bought the land, built houses, and receives rent from the tenants. For some cases, the landlord/homeowner did not live with the tenants. Whether the respondent was a homeowner or tenant, we chose inland residents only.

### 2.4. Sampling

For the KIIs, we collected data on (i) the current sanitation status in LICs, (ii) implementation strategies under DSIP or otherwise, (iii) recommendations for strategies to connect LICs to sewerage, and (iv) future plans for implementation of DSIP. Key persons were selected for conducting key informant interviews from Dhaka Water Supply and Sanitation Authority (DWASA), the Pagla Sewerage Treatment Plant, and Dhaka City Corporation. Following an open-ended questionnaire, we conducted KIIs with the officials who were directly involved with DSIP and were actively engaged with the existing and proposed sewerage treatment plant. We also interviewed representatives from a non-governmental organisation who were involved in urban sewerage service delivery. Key personnel from an Engineering Research Institute, with experience implementing urban sewerage solutions, and DWASA engineers from Maintenance, Operation, Distribution, and Service (MODS) zones involved in strengthening the sanitation infrastructure, were also included. The interviewees were selected based on their experience and knowledge regarding the Dhaka Sewerage Master Plan and those who were closely involved with Dhaka Sanitation Improvement Project. The interviewed participants have been listed in Table 1.

For FGDs, we selected residents of LICs who were either tenants or house owners. We also selected CBO/community leaders who were responsible for monitoring/decision making for LICs utilities or who participated in various implementing activities previously. A rapid visit was carried out to select and finalise the field sites for conducting FGDs for this study. Additionally, information on the number of households, population, toilet type/connection status, and distance from the trunk main was also collected during that visit. Sixteen areas were selected and visited separately from the different low-income communities across Dhaka city. FGDs were conducted with landlords/homeowners, tenants, and community leaders in their own communities to get a communal response and to avoid excluding important perspectives of those who would find travel difficult. 

Twenty-three FGDs were conducted in total. Each discussion was conducted with 6–10 participants. These included mixed groups consisting of (i) house owners/landlords and tenants together, (ii) groups of only house owners/landlords, (iii) groups of tenants with current faecal sludge arrangement to storm drainage, (iv) groups of tenants and house owners without sewerage connection, and (v) a group of community leaders (Table 2). We tried to consider gender, age, and occupation while choosing our FGD participant groups. However, except for tenant groups, in most cases, participants were males. These FGDs assessed the satisfaction of the users regarding sewerage facilities in terms of their financial status in different LICs, the demand for sewerage systems, barriers for implementation, challenges, and opportunities of connecting LICs to a sewerage network, and affordability of proposed policies and strategies.

### 2.5. Data Analysis

Audio recordings of the KIIs and FGDs were transcribed, translated, and the following themes and codes were finally summarised (Appendix A). Conceptually similar data were grouped into sub-categories and subtopics/themes. Summarised data were coded primarily following the inductive reasoning approach, where all collected information was considered. While coding, firstly, open coding was done where concepts were labelled and defined, and categories were developed based on the dimensions of the information. Afterwards, data was related together through axial coding in order to reveal codes and to identify relationships among the open coded data. Finally, core categories/themes were identified through selective coding, which included all the data, and thereby primary findings were illustrated thoroughly based on the major thematic areas. Thus a combined code list was prepared for analysis, and data were then analysed using thematic analysis method [35]. We also cross-checked our data during fieldwork to ensure its validity while talking to different respondents in different areas. During organising, coding, interpretation, and in all stages of data analysis, we did not intend to misinterpret any data by prioritising our viewpoints (Figure 2).

In this study, thematic analysis was chosen not to merely count phrases or words in a text but to explore explicit and implicit meanings within the primary data. In brief, thematic analysis was performed through the process of coding in six phases to create meaningful patterns. These phases were: familiarisation with data, generating initial codes, searching for themes among codes, reviewing themes, defining themes, and preparing the final report [36]. We included codes in the analysis if they met all of the following evaluation criteria:Overlap of a particular code (e.g., “narrow roads” was mentioned with “Major risk and challenges” by the participants);Mentioned in at least two different FGDs;Mentioned in a KII with DWASA personnel engaged with DSIP.

### 2.6. Ethical Approval

We obtained ethical approval from the Ethical Review Committee (ERC) of the International Centre for Diarrhoeal Disease Research, Bangladesh (icddr,b). Study participants were informed of the aims of the study and their rights. Enumerators read an information sheet to respondents in Bengali, answered any questions raised, and obtained written consent for participation. Respondents were given a copy of the information sheet to keep, and no compensation was provided for participation. Names and numbers were removed from final data sets to protect anonymity.

## 3. Results

In the first evaluation step (open coding), based on our evaluation criteria, we identified 5 core themes comprising 9 categories and 19 sub-categories (Appendix A). However, after analysing the qualitative findings, the two codes that reached the highest score were: “government should build a sewerage network” and “government should take the responsibility to monitor it.”

### 3.1. Perceptions of Current Faecal Sludge Management Situations

Only one group out of 16 study areas had private household toilets, while 13 FGDs mentioned “shared toilet” as their only toilet facilities. These respondents typically shared their toilets with 5–30 other households. Toilets were almost always provided by “different NGOs” (15 FGDs). Only 3 areas had septic tanks connected to their toilets, but they were not connected to the main sewerage line. “Narrow lanes” were one of the most commonly reported problems with the current sewerage system in 18 out of 23 FGDs. Collection and management of faecal sludge were mentioned as difficult to carry out either manually or via vacutag in those areas because of these narrow lanes. Exit pipes of toilets were usually “connected to storm drainage.” Among 16 LICs, we found storm drains completely open in 3 LICs, yet several toilet exit pipes were connected with those drains. In 11 LICs, storm drains were mostly covered but still had minor uncovered sections. We did not find any storm drainage in the rest of the LICs. These types of connections eventually allow faecal matter to be disposed into an open water body near to the study areas such as a canal or a lake. Moreover, three different additional “technical issues” were identified. These were “pipe blockage,” “pipe leakage,” and “narrow connecting pipes” (Appendix A). “Pipe blockage” was mentioned 51 times, “pipe leakage” was mentioned 36 times, and “narrow connecting pipes” was mentioned 17 times with “problems of the current sewerage system.” Nine tenant FGDs had mentioned that during the rainy season, there was frequent overflow inside their household area from the drains and canals where faecal matters finally exit. They perceived that it occurred due to insufficient drainage facilities for disposal of faecal material and other wastes during heavy rainfall. Both tenant and homeowner groups in 2 FGDs mentioned that their houses were situated at a lower elevation than the nearby sewer pipes, which resulted in frequent overflow of wastes after heavy rainfall during the monsoon season.

“*In the rainy season, our excreta travel back due to an overflow in the drains; it is common that dirty water with faecal matter enters even into our living room.*”—FGD, Female tenant, IG Gate Bank Colony

Furthermore, “no water supply facility inside the toilet” was mentioned by the participants 62 times with “problems of the current sewerage system.” Hereby, respondents addressed that there was no water supply line inside the toilet and as such whenever they used the toilet, they needed to take water from external water sources (such as nearby tube-wells) for flushing and washing. They mentioned it as a severe problem as it is difficult to take sufficient water along with them when they use the toilet. Therefore, they cannot properly practice sanitation-related personal hygiene practices or clean the toilet. Landlord/homeowner groups from the four study locations out of sixteen mentioned that the land on which they were residing was owned by the government, and it was repeated in 3 FGDs that they did not have legal registration from the government. Therefore, they remained uncertain about the government granting permission to them to live there, and hence were not taking any initiatives to improve their toilets. In these three study areas, it was visible that faecal matter and household wastewater were openly mixed with a nearby water body.

“*Most of the toilets in our slum are hanging toilets, and slabs are set upon the bamboo-made floor, and faeces finally go to the open water body, even our houses stand upon the water body.*”—FGD, Male tenant, Mogbazar slum

Four different “health hazards” commonly met the evaluation criteria. They were “bad odour,” “cholera,” “diarrhoea,” and “skin diseases” (Appendix A). “Bad odour” was mentioned 113 times, “diarrhoea” was mentioned 72 times, “cholera” was mentioned 43 times and, “skin disease” coincided 39 times with common “health hazards” because of the situation of the current sewerage system. Tenant groups in 2 FGDs had experienced faecal odour in their supply water.

“*Ours is a hanging toilet. We don’t even have any drain here. So faecal matters stuffed beneath the ground of the toilet and travel with the water we use in the toilet*”.—FGD, Male house owner, Tekpara

### 3.2. Perceived Benefits of Sewer Connection

Two broad themes met the evaluation criteria through which a range of benefits was identified from all the FGDs. These were environmental benefits and health benefits. Based on all FGDs with landlords/homeowners, tenants, and community leaders, the most frequently mentioned environmental benefits of having a sewerage connection were safe disposal, no bad odour, no clogged drain, no overflow, safe drinking water, and no contact with wastewater. In 19 FGDs, participants perceived that having a proper sewerage network would be the safest system, and the enclosed disposal of faeces would prevent the openly passing raw excreta. Participants expected that if their existing toilets can be connected with the proposed sewerage network of DWASA, it would be safer for them in terms of their health and hygiene practices. Generally, in all study areas, all the tenants of each group expressed a stronger desire to have a sewerage connection than landlords, mentioning that it would result in a cleaner environment quality as well as prevent various diseases. Moreover, sewerage connections were seen as a way to prevent bad odour, which was the most common complaint in 20 FGDs.

Furthermore, tenant groups in 3 FGDs claimed that cleaning the drain was expensive, and no one wanted to take the responsibility of cleaning the clogged faeces in the blocked or leaking drains. Residents wanted to avoid providing cleaning costs or taking responsibility upon themselves. This sometimes resulted in quarrels between neighbours. These groups also hoped that having proper sewerage connections would improve the social relationship among the neighbours.

Another major environmental benefit perceived by the tenants in 5 FGDs was that wastewater would no longer overflow onto their surroundings, and their drinking water and food would not be contaminated by it. They perceived that drinking water pipes would not mix with the drain water pipes, if a proper sewerage connection can be installed. In 2 FGDs with tenant groups, it was also mentioned that wastewater contact with their skin could be avoided, and Muslims would not have to worry about becoming impure if the faecal matter did not travel back and overflow into their surroundings.

“*We are living by the side of a canal. This canal is badly filled with huge wastes. All its need is 10 minutes of heavy rainfall to overflow of waste in our surroundings. We could have separate drains to pass different wastes in a proper way. If the sewages don’t mix with other wastes and don’t go to the canal, there is a less chance to see them back with the overflow*”.—FGD, Male tenant, Khilgaon Bagicha

Based on all FGDs with landlords/homeowners, tenants and, community leaders, the most frequently mentioned health benefits of having a sewerage connection were no contact with pathogens, no contamination, no mosquitoes and flies, no breathing problems, and no skin diseases. In 21 out of 23 FGDs, participants perceived that having a proper sewerage network would eventually lead to better health of their children as well as the general public. Having a cleaner and safer disposal of faeces would lead to the prevention of contact with disease-causing pathogens and lesser incidences of diarrhoea, cholera, jaundice, etc. Moreover, if the excreta did not end up in water bodies, their drinking water and food would not be contaminated, and they could avoid many waterborne diseases. Along with these health consequences, respondents in 5 FGDs with tenants and 6 FGDs with landlord/homeowners also hypothesised that this proposed sewerage network would also have a great positive impact on the environment. They perceived that if the LICs would be connected with the proposed sewerage network, both environmental hazards and poor health consequences will be lessened. In 8 FGDs, both landlord/homeowners and tenant groups expressed that mosquitoes and flies would not have emerged if the water bodies were clean, and thus dengue and chikungunya could also have been avoided. Tenant groups expressed greater concern about possible health benefits. In 6 out of 10 FGDs with the tenant groups, respondents hoped that they would not have to visit the hospital too frequently, and health-related costs would be reduced. They asserted that this proposed sewerage connection is important for them to mitigate the negative consequences of current faecal sludge arrangement to storm drainage in terms of their health crisis. Due to their limited income and demographic situation, such proposed services would be beneficial for them, which they could not afford by themselves.

### 3.3. Willingness to Connect to a Sewerage Network

Three key personnel from DWASA, who are also actively engaged in DSIP, mentioned that households within 100 feet of the proposed main sewerage line would be forced to connect with that sewerage network after imposing regulations by concerned authority (i.e., City Corporation or DWASA). They also added that residents of these areas might need to obey this regulation in order to continue their residency. This plan was also evident in the Sewerage Master Plan documents. Four different technical strategies met the evaluation criteria with “possible strategies” (Appendix A). These were “building a sewerage network on government’s cost” (4 KIIs), “improve existing toilets,” including hanging toilets (2 KIIs), “subsidised service charges” (2 KIIs), and “providing loans” (2 KIIs). One KII mentioned “Providing loans” to the homeowner for improving their toilet to connect with the sewerage network to increase sewerage connection rates.

Participants in almost all the FGDs were found to be willing to connect their toilets with a sewerage network under some circumstances, with five key conditions mentioned—“no installation cost“for sewerage connection, “household type/size based service charges,” “area-based subsidies,” “income-based subsidies,” and “financial support for toilet improvement” (Appendix A). “No installation cost” was mentioned 77 times. Landlords/homeowners had mentioned that the government should bear the cost of building a sewerage network in 7 FGDs while offering that they could contribute a tiny part for the connecting pipes to connect their toilets with the network. “Household type/size based service charges” was mentioned 17 times in 2 FGDs with tenant groups. These groups perceived that the monthly service charge should be fixed based on the toilet type and the number of users. “Area-based subsidies” was mentioned 23 times, and “income-based subsidies” was mentioned 31 times along with “willingness to pay.” These subsidy types were mentioned in one FGD of community leaders and 4 FGDs with tenant. Landlord/homeowner groups in 3 FGDs and tenants in 2 FGDs disagreed about the best modality, with some claiming that service charges should be fixed in terms of toilet types and the number of users in a single-family because they share their toilet with other families. Other respondents stated that as they live in a low-income area (i.e., Mohajer colony) and they earn poor wages in comparison with the other areas of the city, subsidies should be fixed considering the type of area. The latter group perceived that it would be better if they had to pay subsidised service charges since they were from a low-income community and were living in poor conditions. In 3 FGDs with landlord groups, “financial support for toilet improvement” was mentioned 12 times. Financial support from the government or other non-government organisations was considered necessary if the costs of renovating existing toilets to be able to connect to sewers were too high. Homeowners living on government land wanted to ensure their investment would not be lost due to displacement. They were not informed whether they would get permission from the government to stay. These respondents commonly mentioned that, in the recent past, several LICs who used to live in a government land got a legal notice to leave their residences. Therefore, currently, they are also afraid of facing such cases. For this reason, they were not willing to invest much in sanitation.

“*In the case of water lines, the number of plots in each street was counted, and an underground connection was created with each plot so that no matter what, households would be able to connect and future expansion would be possible with the pre-developed system. A similar strategy will be followed for the sewerage line so that the network is present for users to connect. And also, the LICs in the city which meet these criteria will be able to connect. Proper community mobilisation is required for increasing sewerage connection number among these LICs.*”—KII, Key personnel, DWASA

The DWASA Sewerage Master Plan of Dhaka City is designed to ensure that sewerage facilities are accessible to the whole city. Only a few technical strategies met the evaluation criteria with “possible strategies,” such as a “communal tank,” which was mentioned in 4 KIIs. Relevant DWASA authorities have an initial plan to build communal septic tanks (underground) in some areas where placing connecting pipes is almost impossible due to narrow lanes within a community. Stacked wastes would be transferred and disposed into the Pagla Sewerage Treatment Plant (PSTP) station via the main trunk line. However, the lack of sufficient space to set up such communal septic tanks was mentioned in 2 FGDs with landlord groups. A few additional technical strategies were suggested in the 4 KIIs with DWASA officials. Frequently suggested strategies included:Imposing residency regulations upon community members to get connected with the sewerage network;Rebuilding the hanging toilets in LICs by DWASA authority;Allowing communal toilets for LICs to be built under DSIP;Dhaka City Corporation should widen the roads and operate on-site sanitation in difficult-to-connect areas.

### 3.4. Affordability of Having Sewerage Connection

We explored to what extent users are willing to pay for sewerage network connections and maintenance during the focus group discussions. Preferences regarding payment methods and affordability varied primarily based on their financial situation and occupancy status as a landlord, house owner, or tenant. In most cases, the participants were willing to pay for their sewerage connections as they believed that it would benefit them. Four different preferences met the evaluation criteria with a possible payment method. These were monthly bills, one-time payment, equated monthly instalments (EMI), and cash vouchers. Monthly bills were mentioned 68 times with possible payment methods for sewerage service charges. Tenant groups in 8 FGDs mentioned that similar to other utility bills, and they could pay a monthly service charge.

It has been mentioned in the sewerage master plan that sewerage service fees may be charged up to 2% of the total monthly income of the users based on the services received. Based on all the FGDs, the amount ranges from 50–500 taka monthly. Among 8 LICs, tenants from Kamalapur Railway Colony, Kamrangirchar, and Duaripara wanted to pay within the range of 50–100 BDT, while tenants from Tekpara and Dholpur mentioned a range of 200–400 BDT, and the tenants from Lalchanmukim Lane, Maniknagar, and Nobinbag could afford 300–500 BDT as sewerage service fees. This amount would be paid by dividing the charge among the user households. This range was regarded as “affordable” based on all FGDs with tenant groups. Nevertheless, these 8 tenant groups were not at all willing to pay for sewerage connection installation or toilet improvement costs. Moreover, they perceived that it was the responsibility of the landlords to manage sewerage facilities to make the lives of their tenants easier. In their existing situation, the tenant groups were bearing the cost for minor repairs and regular maintenance; the amount ranges from 50–500 taka per household depending on the toilet condition. In 2 FGDs, tenants reported that they were tackling minor repairs communally, but the landlords/homeowners should provide adequate toilet facilities.

We found tenant groups paid a certain water bill (200–300 BDT) monthly to the water suppliers, such as the water pump owner, from whom they collected their drinking water. Two tenant’s groups in Lalchanmukim Lane and Shyampur were found buying water from mosques by paying a fee per litre. Tenants in only one study area used a water supply provided by an NGO for which they had to pay 800 BDT (9.4 USD) per month on average. One-time payment was mentioned 13 times with possible payment methods for sewerage connection installation in 4 FGDs with the landlord groups. In most cases, they preferred their “affordable” range, which is from 10–20% of the total cost needed to have sewerage connections in their toilets. This percentage is perceived by them as affordable based on their income and previous experiences like sharing costs among themselves while setting up a toilet by different NGOs, considering the context of IG Gate Bank Colony and Maniknagar slum where few toilets were built by different NGOs and homeowners of these areas bore the total cost through EMI as well as sharing costs among themselves. As such, in this study, they perceived the 10-20% range of the total cost as affordable for them. Nevertheless, they demanded financial help from the government to manage the rest of the amount.

Equal monthly instalments (EMI) was mentioned by the participant 7 times, and cash vouchers was mentioned 4 times with possible payment methods for sewerage connection installation and toilet renovation cost by the landlord/homeowner and community leader groups. In 3 FGDs, landlord/homeowner groups suggested that they could afford EMIs for a certain period if the government or NGOs initially rebuilt their toilets with a proper sewerage connection. Community leaders in 2 FGDs also suggested that being a low-income community, respective homeowners would be able to afford the least amount. The government could provide financial support like cash vouchers, discounts, and easy loans to renovate their toilets. Apart from this issue, landlords/house owners in at least 4 FGDs repeated that some of them might not be able to afford the required amount for both sewerage connection and toilet renovations or setting up new toilets. Those whose households were far away from the proposed main sewerage line may require longer connecting pipes as well as more money to get connected to sewer networks than others.

“*We are poor in terms of our income, and it becomes difficult for us to afford all of our basic needs. We cannot afford a high amount of money regarding toilets. If the government pays ¾ of the total cost, then we will pay the rest ¼ for toilet improvement.*”—FGD, Male CBO leader, Dhalpur

In 2 KIIs, participants hoped DWASA would execute a plan to install sewerage connections free of cost at the LICs. At present, tariffs are equal for all types of DWASA consumers. In 3 KIIs, it was assumed that if the government permits, tariffs could be reduced for LIC residents. DWASA may also introduce a certain level of cross-subsidies for sewerage bills, although such options have not yet been explored in detail.

### 3.5. Barriers of Being Connected with the Sewerage Network

The core limitation of the existing sewerage network of Dhaka city is that it only covers 20% of the total area. The Sewerage Master Plan aims to connect the whole city under a single sewerage network, which is to be completed by the year 2035. However, there are several barriers which may prevent this.

Based on 9 KIIs, the most frequently mentioned barriers under major risks and challenges were ever-growing populations, high-rise buildings, narrow roads, overlapping of various utility connections under the same road, and an old sewerage network (Appendix A). Densely populated areas in slum settlements are likely to be one of the major barriers to the implementation of the DWASA strategy. Along with this population density issue, one KII indicated that previously it was possible to install a sewerage pipe below 150 centimetres in diameter. However, the growing numbers of high-rise buildings and population density require it to be a minimum diameter of 200 centimetres, which is challenging for the implementing authority. Overlapping of various utility distribution lines such as gas, water, or other utility lines passing through the manholes were also found as major obstacles for annual mass cleaning.

“*Changing and replacing pipelines at a time running under the city is not possible since the entire city would face heavy traffic. Because all pipelines are installed underground of the road, and if these lines need to be repaired or replaced, roads will be blocked*.”—KII, Executive Engineer, Sewer Division, DWASA

Moreover, the existing sewerage network is too old to function. Leakages and waste overflow were frequently reported complaints. Narrow connecting roads in some areas like the old Dhaka region make cleaning activities very difficult even with a vacuum truck.

“*Many roads and lanes in Old Dhaka areas are extremely narrow, and setting up a new sewerage line will be a tough task and almost impossible there*.”—KII, Key informant, DWASA

At present, the only sewerage treatment plant at Pagla (PSTP) has a capacity to treat 120 mL/d per day, whereas only 50–70 mL/d is being brought into the plant. This happens because secondary and tertiary lines have blockages in different locations since the network was built in 1977.

“*The pipe of the new trunk main will be more than 5 feet in diameters. However, in some areas, the secondary and tertiary pipe’s diameter is about 2 or 3 feet, which is narrower than the requirement of that area. This narrow sewerage pipes often get clogged with other waste that enters during various construction works*.”—KII, Key informant, DWASA

## 4. Discussion

Ensuring sewerage coverage for all residents in Dhaka city will be challenging for the Dhaka Sanitation Improvement Project (DSIP) due to the many engineering and management barriers. This study explored the challenges and opportunities of connecting LICs to a sewerage network and the affordability of connecting those LICs to a proposed sewerage system from a financial and infrastructural perspective. In general, the study confirms previous findings on the challenges facing governments and utilities to provide adequate faecal sludge management in slum settings. Furthermore, it provides more detailed information on the perceived barrier and facilitators providing sewers in LIC settings to inform strategies to achieve sewer connections for residents of dense low-income urban settlements.

Solid waste is a critical issue in slums, especially the more congested ones [37]. In this study, drain pathways and water bodies were also invariably reported to be filled with faecal sludge. It was evident from the observation that there is a lack of sanitation infrastructure in every visited LIC, and most of the population had pour-flush sanitation systems and do not utilise septic tanks. Hence, untreated faecal wastes are openly disposed of in canals leading to the risk of groundwater infiltration, with potentially severe consequences on human health and physical environment [38]. Collection and management of faecal sludge were difficult to carry out because of the narrow lanes within LICs. Currently, residents of the LICs share the cost of emptying the tanks. Respondents preferred sewerage connections rather than having a septic tank as septic tanks were perceived to cost more than sewerage connections initially. Moreover, infrastructural limitations like the absence of water supply, drainpipe blockage, and leakage, and narrow connecting drainpipes met the evaluation criteria for issues with the current faecal sludge arrangement to storm drainage. Bad odour, cholera, diarrhoea, and skin diseases were also identified as the most negative effects of current poor sewerage facilities. LIC residents argued that they would prefer to connect their toilets with the proposed sewerage network to lessen their existing physical and environmental hazards.

For improving the current situation of sewerage facilities, DWASA has planned to fund and build a sewerage network; improve existing toilets including hanging toilets; subsidise service charges for the poor; provide loans for toilet renovation or for building new toilets, and for installing sewerage connection pipes; and build communal septic tanks for areas mentioned in the Sewerage Master Plan [26]. Four suggestions were recommended by the landlords/homeowners and tenants, including fully subsidising the installation cost for sewerage connections, collecting service charges based on household type/size, providing area-based subsidies and income-based subsidies, and providing financial support for toilet improvement. A similar model exists in another country- a simplified sewerage model was deployed in the Orangi Pilot Project (OPP) in Pakistan, with financing mechanisms divided between the residents and the government. In such case, the residents would finance the smaller scale internal components such as household sanitary latrines and underground sewers, whereas the larger trunk main external sewerage installation finances were borne by the local government. Residents of a particular lane were considered a single organisational unit allowing internal funding to be cohesive among households [39].

Notably, financial support from the government for building sewerage connections and subsidised sewerage service charge was the most prioritised strategies suggested by the participants. These two were closely related to the level of affordability of the LIC users. A community’s average cost of water and wastewater services within a municipality’s district is measured as a percentage of the median household income within the city limits. If this value is greater than 4.0% or 4.5% for both water and wastewater services, the system is considered to be of high cost and not affordable for families [40]. Consistent with these affordability parameters, our study findings confirm four different preferences, which were commonly identified as a possible payment method. These were monthly bills, one-time payment, equated monthly instalments (EMI), and cash vouchers. The tenants stated that landlords should bear the installation and maintenance cost of the sewerage network, although they were willing to pay a share of the total costs via the monthly bill. Landlords agreed to pay a maximum of 20% of the total installation cost of sewerage connection from trunk main to the LIC toilets. For setting up the septic tank and installing connecting pipes from the proposed trunk main, providing loans to landlords was suggested by a few key personnel of DWASA and landlord groups of the study. Though NGOs tried to meet people’s sanitation needs in different ways [41], past studies found that LICs often are excluded from sanitation programs implemented by both the government and NGOs as well [42,43]. Moreover, it was largely neglected by WASH NGOs that land tenure/insecurity may potentially impact on the successful implementation of a sanitation project [44]. The present study noted that among 16 LICs, a significant portion of the participants who are living in a government land without legal permission expressed concern that their permanent residency must first be ensured before they would consider sharing in the resources needed to be connected to the sewerage network. This was consistent with the results from another study for slum upgrading in Bangladesh [16]. Furthermore, a study in low-income urban areas in Senegal also found that tenants were more willing to pay for operational costs rather than capital costs of sanitation due to having limited security for tenure [15], thus exemplifying the fact the permanent residency status affects the decision to invest in household sanitation. Although DWASA Sewerage Master Plan is designed to ensure sewerage facilities for the whole city, there is no clear decision about legalising illegal residences. Residents of LICs noted that the ever-present danger of eviction is particularly threatening for those illegally occupying public lands. Other slum dwellers who were renting space were also more or less vulnerable regarding eviction, consistent with previous studies [45].

While introducing the proposed sewerage network in our study LICs, the issue of the illegal settlement was also addressed as one of the major problems behind the current state of sanitation. Participants from 8 LICs asserted that as they were not permanently settled, they were not willing to renovate the existing toilet facilities or even interested to bear any cost of installing the sewerage connection. Therefore, this study tried to identify their interest to pay for being connected to the proposed sewerage network within their affordability range, which has remained unaddressed in previous studies on this issue [46].

Present findings noted that by overcoming the stated barriers, most of the proposed policies to connect LICs with the sewer network are feasible given strong demand from residents themselves. As seen by the OPP’s approach to sanitation, most of Orangi’s informal households and settlements in Karachi had adopted the simplified model of the sewer program overcoming collective action, co-production, affordability, and technical challenges [47]. Growing populations, high-rise buildings, densely populated areas in Dhaka’s slum settlements, narrow lanes, overlapping of various utility connections under the same road, and an old sewerage network were the prime barriers. Despite existing demand for improved sanitation, there remains limited scope for these slum dwellers to improve their sanitation conditions. To achieve the ultimate outcome of the DSIP and to ensure proper connections, community mobilisation efforts must be initiated by DWASA and include capabilities beyond their existing ones, perhaps enlisting the aid of NGOs or other organisations or creating distinct business units within the utility [48].

## 5. Limitations

Our study was limited to LICs near the proposed Eastern Trunk main, and thus may not be representative of all of Dhaka or generalisable to other settings. However, the current study covers most potential LICs relevant to the present phase of DSIP and forms the basis of further assessments and evaluations, which may provide more generalisable learning. Secondly, a number of priorities were reported by the FGD participants, but this study did not consider the associated costs of each priority. As policymakers need to understand these trade-offs before making large scale decisions, these findings should be used to inform large-scale quantitative surveys, as were conducted subsequent to this qualitative phase.

## 6. Conclusion and Recommendations

Our study sought to represent the voice of LIC residents on the issue of faecal sludge management regarding both preferred services and their view of the consequences of an improper sewerage system. This study captured a range of situations across 16 low-income communities, with an emphasis on understanding the socio-cultural context. Through customised and context-oriented plans, there is reason to expect that LICs can be connected with the main sewerage network. Residents of the LICs currently lack high-quality sanitation facilities and as such, they expressed their need to have sewerage connections for ensuring better living conditions. Key recommendations for policymakers generated from the findings of the study were as follows:The perceived necessity of providing a cost-free installation of sewerage connection;A need to review the legal framework for residency/land tenure for the unregistered LICs;Ensuring that the utility/service providers adequately conduct community mobilisation;Ensuring financial support both from governmental and non-governmental organisations;Setting affordable service fees for the users;Introducing appropriate subsidy structures, including income- and area-based subsidies, for tariffs;Deploying alternative sewerage treatment procedures, where necessary;Imposing strict laws to reduce faecal waste disposal to open water bodies.

For estimating cost and user willingness-to-pay for different sewerage arrangements, a quantitative study is needed. In addition, the potential role of a designated community member for the maintenance of sewerage facilities at the community level should also be investigated. It may be feasible for DWASA to potentially introduce a single model (utility tariff collection process similar to electricity bill) to manage a sustainable sewerage service for all.

## Figures and Tables

**Figure 1 ijerph-17-07201-f001:**
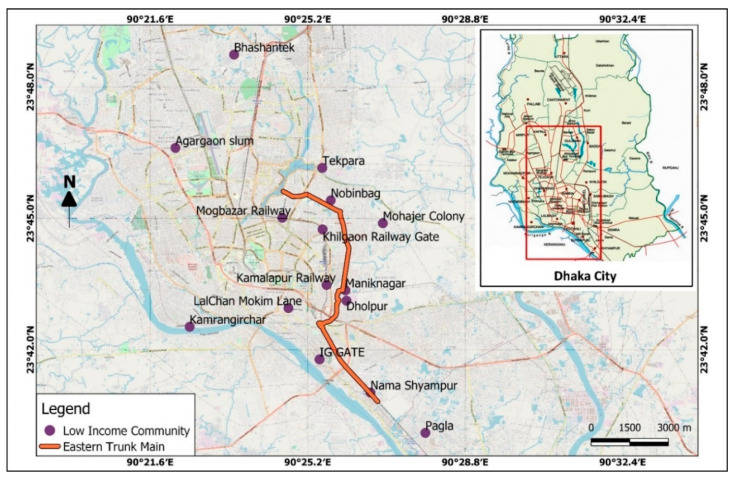
Study location of the low-income communities (LICs) with proposed trunk main.

**Figure 2 ijerph-17-07201-f002:**
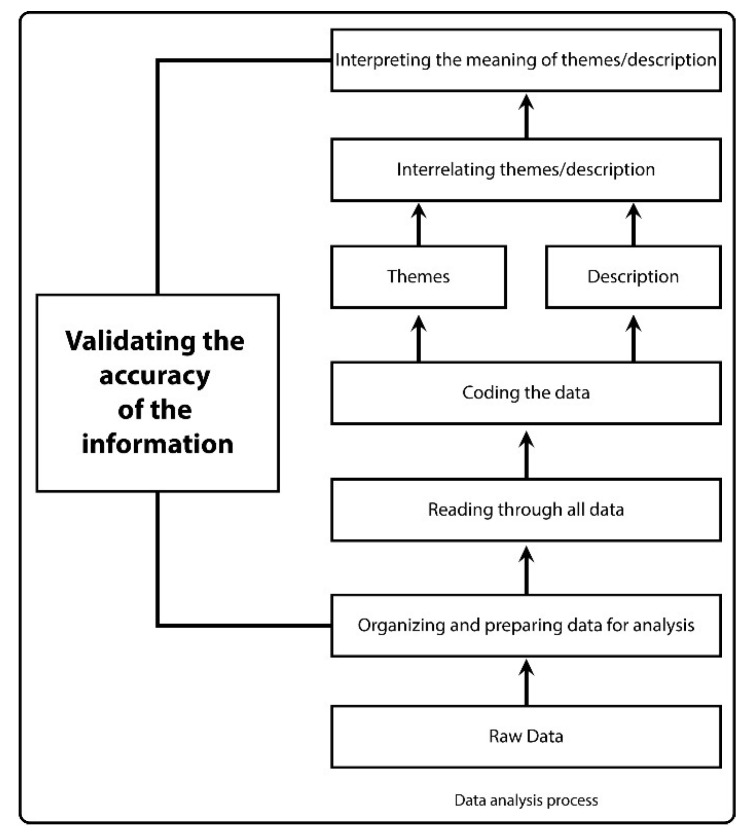
Data analysis process for the thematic analysis method.

**Table 1 ijerph-17-07201-t001:** Key informant interviews with the government and community stakeholders.

Designation	Organisation/Department
Executive Engineer	(C. C.) DWASA
Executive Engineer	P&D (Sewer) Division of DWASA
Senior Community Officer	DWASA
Executive Engineer	Pagla Sewerage Treatment Plant (PSTP)
Ward Councilor and Board member of DWASA	Councillor of 26 no. Ward Dhaka City Corporation and Board member of DWASA
Executive Engineer (2)	MODS Zone (Jatarbari service area), DWASA
Research Officer	ITN-BUET
Deputy Director	Dushtha Shasthya Kendra (DSK)

**Table 2 ijerph-17-07201-t002:** Number of focus group discussions (FGDs) with slum residents.

Group of participants	Number of FGDs
Landlords	9
Tenants with sewer connection to the storm drainage	4
Tenants without a sewer connection	6
Landlord-tenant (mixed)	2
CBO leaders	2

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
