# Peer review of "Strategies to Connect Low-Income Communities with the Proposed Sewerage Network of the Dhaka Sanitation Improvement Project, Bangladesh: A Qualitative Assessment of the Perspectives of Stakeholders"

_ijerph, 2020, doi:10.3390/ijerph17197201_

Round 1

Author Response

Response to reviewer # 1 comments

Main Points

Contribution & References

  1. Emphasizing your specific contribution and originality (currently on line 89-91) is recommended earlier in the introduction, and throughout the paper.

Response: Thank you so much for your suggestions. We added sentences (currently 96-102) to point out two important lacks of sewerage master plan.

“However, this master plan does not clarify how connections to sewerage network will be made for illegal settlements. A clearer illustration of financial support for those not being able to afford toilet renovation was also absent in the plan.”.

  1. Engaging with other literature on sewerage and LICs both focusing on, and beyond Bangladesh may be beneficial, to demonstrate that solutions are possible. For example, the simplified sewerage model deployed in the Orangi Pilot Project (OPP) in Pakistan (for details, see: McGranahan, G., and D. Mitlin (2016) ‘Learning from Sustained Success: How Community-Driven Initiatives to Improve Urban Sanitation Can Meet the Challenges’, World Development 87:307–317 and Hasan, A. 2008. Financing the sanitation programmes of the Orangi Pilot Project (OPP)-Research and Training Institute in Pakistan. Environment and Urbanization, 20 (1): 109–119). Relevant studies that discuss WTP (including among tenants) and sanitation also include: Scott, Pippa, Cotton, Andrew & Khan, M Sohail (2013). Tenure security and household investment decisions for urban sanitation: The case of Dakar, Senegal. Habitat International, 40, 58-64.

Response: Thank you so much for your suggestions. We have tried to show a connection between existing literature and our study. We have added in Line 78-79:

“Additionally, low income households usually refrain from investments in sanitation due to tenure insecurity”

We have added in Line 592-598:

“A similar model exists in another country- a simplified sewerage model was deployed in the Orangi Pilot Project (OPP) in Pakistan with financing mechanisms divided between the residents and the government. In such case, the residents would finance the smaller scale internal components like household sanitary latrines and underground sewers whereas the larger trunk main external sewerage installation finances were bore by the local government. Residents of a particular lane were considered a single organisational unit allowing internal funding to be cohesive among households.”

  1. One striking absence is the mention of the change in DWASA’s citizen charter to provide legal water connections to LICs regardless of tenure/holding numbers, and the establishment of the LIC department. This may demonstrate to readers that legal service connections can and have been rolled out to LICs in Dhaka and that residents have contributed to this in cash and kind (though not without challenges, and with incentives for DWASA oriented towards cost-recovery and revenue collection, not necessarily the human right to water for all). See: Hossain, K. Z. and S.A. Ahmed (2014) ‘Non-conventional public- private partnerships for water supply to urban slums’, Urban Water Journal 12: 1–11; Akbar, H.M.D. Minnery, J.R. van Horen, B and Smith, P. 2007. Community water supply for the urban poor in developing countries: The case of Dhaka, Bangladesh. Habitat International, 31 (1): 24–35; Rana, M.M.P. and Piracha, A. (2018), "Supplying water to the urban poor: Processes and challenges of community-based water governance in Dhaka city", Management of Environmental Quality, Vol. 29 No. 4, pp. 608-622 and Rana, M.M.P. and Piracha, A. (2020). Supplying water to the urban poor: Government’s roles and challenges of participatory water governance. Cities, 106 (to name a few).

Response: Thank you so much for your suggestions. We added in line 116-122:

“In order to provide legal water connections to several areas in Dhaka including LICs regardless of tenure/holding numbers, DWASA has previously altered and updated its citizen charter. For such cases, a LIC department was also established, although informal markets continues to play a crucial role. Since legal service connections have been extended towards LICs in the past with resident’s contributions via incentives directed for cost-recovery and revenue collection for DWASA, such strategies can be adopted for the case of providing legal sanitation services to the low income residents as well.”

  1. Line 102-103 – the authors refer to ‘many studies’ on water and pricing but only cite 1 (reference 20) – are there others that can be included here? For example: Muhammad Mizanur Rahaman, Tahmid Saif Ahmed. Affordable Water Pricing for Slums Dwellers in Dhaka Metropolitan Area: The Case of Three Slums. Journal of Water Resource Engineering and Management. 2016; 3(1): 15–33p; Haque M. (2019) Urban Water Governance: Pricing of Water for the Slum Dwellers of Dhaka Metropolis. In: Ray B., Shaw R. (eds) Urban Drought. Disaster Risk Reduction (Methods, Approaches and Practices). Springer, Singapore. and BIGD 2019. State of Cities 2018: Water

Response: Thank you so much for finding these two literatures for us. We found their relevancy and added accordingly in Line 135.

 Housing and Land Tenure Insecurity

  1. Whilst the authors quite rightly highlight the importance of land tenure insecurity, especially at the end of the paper (lines 342-344), I feel it requires a more central positioning earlier on and in the methodology (see below). Clarification is also needed around ‘landlords’ and tenants. When you [the authors] state ‘landlords’, do you mean owners of the land who live in the settlements, or house owners? Or, are you referring to landlords living outside of the settlements? Some landlords and tenants have also been found to ‘block’ NGO or government infrastructure investments, due to fear of rising rent and bills. House owners and tenants also mention land grabbing and encroachment by elites (including businesspersons and politicians) after improvements – Did these issues emerge?

Response: Thank you so much for raising an important issue. We have added in Line 78 and 79 about tenure:

“Additionally, low income households usually refrain from investments in sanitation due to tenure insecurity.”

Under the section 2.3 (Line 197-200) in methodology we added clarification to clear-up ‘landlord’ concept for the study-

“In the study, the group ‘landlord’ means he/she owned the house only, or they were allocated to use the land for a certain time, or who bought the land, build houses and receiving rentals from the tenants. In some cases, landlord/homeowner was not living with the tenants. Whether the respondent was a homeowner or tenant, we chose inland residents only.”. 

  1. For details, see: Banks, N., M. Roy and D. Hulme (2011) ‘Neglecting the urban poor in Bangladesh: research, policy and action in the context of climate change’, Environment and Urbanization 23(2): 487–502; Roy, M., and D. Hulme, D (2013) ‘How the Private Sector Meets the Demand for Low-Income Shelter in Bangladesh’, Shelter 14(1): 90–100.

Response: We found relevancy and added those references. In line 616-617 -

“Moreover, it was largely neglected by WASH NGOs that land tenure/ insecurity may potentially impact on the successful implementation of a sanitation project”

  1. Land tenure in/security also remains a largely neglected issue by WASH NGOs who see it as outside of their mandate or too contentious. For details, see: Habib, E. 2009. The role of government and NGOs in slum development: the case of Dhaka City. Development in Practice, 19 (2): 259–265 and Cawood, S., Water and Sanitation in Dhaka's Low-Income Settlements, Bangladesh. 2018, MicroSave.

Response: We found relevancy and added the reference. In line 616-617 –

“Moreover, it was largely neglected by WASH NGOs that land tenure/ insecurity may potentially impact on the successful implementation of a sanitation project.”

Methodology

A number of questions emerged relating to the methodology and analysis, in particular:

  1. Whilst systematic analysis of qualitative data is of course needed, I am concerned that seeking consistent codes across the FGDs and KIIs (line 197-199) may also omit some of the key differences in opinion between the FGD and KII participants. It is no surprise, for example, that opinions (especially relating to solutions, who is responsible or should foot the bill) of the KII participants (with the exception of the NGO) differs from those of LIC residents. This makes me wonder: 1) whose ‘suggestions’ are actually included as recommendations in the end (noting that ‘communal tanks’ – line 345 – are actually only mentioned by the KII participants, not in the FGDs), and 2) whose interests these suggestions actually serve – LIC residents, DWASA, NGOs or all? Greater clarity on this, and how you measured or judged ‘benefits, feasibility and barriers’ (lines 109-113) would be helpful.

Response: Thank you so much for this comment. Actually we have considered a code as consistent if that showed in at least two FGDs. Codes in KIIs were always considered in analysis even if that appeared only once. Therefore, in recommendations we considered everyone’s interests – LIC residents, DWASA, NGOs. We measured or judged ‘benefits, feasibility and barriers’ based on participant’s reported data. However, we have dropped feasibility from the manuscript and replaced with challenges and opportunities.

  1. You also mention (line 204) that ‘no category was consistently mentioned’ (supporting the point above), so is the statement on line 197-199 required?

Response: Thank you so much for this comment. We have deleted this sentence as this creates confusion.

  1. Further detail on the LICs selection would be helpful, specifically:
    1. As land is so central, what were the land tenure arrangements of the 16 LICs? Were they on public, private or disputed land? What impact did this have (if any) on the discussions and suggestions proposed? You mention ‘illegality’, but this is not the same across all settlements, especially the ‘colony’ or ‘staff quarters’ areas which may have partial recognition.

Response: Thank you so much for identifying such an important point. We added few lines to clarify land tenure under section 2.1 (Lines 168-172):

“Among these 16 LICs, 7 were situated on government owned land, 8 were on private land or on land which was leased by a businessman or political person, and the rest was residing on disputed land. Not all of the 7 LICs on government owned land had legal permission to extend their settlements. ‘IG Gate Bank Colony’ and ‘Maniknagar Adorsho Staff quarter’ areas had partial recognition to a certain level. ”

  1. Why did you focus on settlements with less than 500 and more than 1000 households? Economies of scale? What about the many other scattered smaller settlements in Dhaka? Indeed, the BBS survey was also widely criticised for ‘missing’ settlements.

Response: Thank you so much for identifying such an important point. We added few lines under section 2.1 (Line 150-157):

“Areas planned to cover through current phase are located in Dhaka South City Corporation (DSCC), therefore most of the study LICs were selected from DSCC. Additionally, we also chose two LICs from Dhaka North City Corporation (DNCC) to illustrate a comparison with the LICs located in DSCC. Hence, we selected study locations which were located within a distance of 2-3 km from the Eastern trunk main. The study carefully considered distance of the settlements from the proposed trunk main since lack of detailed information regarding user perspectives and way-out to connect, nearby LICs should not be skipped from having sewerage connection”.

  1. Did you take into consideration the disproportionate location of LICs in Dhaka between DNCC (where more are located) and DSCC, during site selection?

Response: Thank you so much for identifying such an important point. We have selected study sites based on the distance from proposed eastern trunk main, so most of the sites were from DSCC. We added few lines under section 2.1 (Line 150-157):

“Areas planned to cover through current phase are located in Dhaka South City Corporation (DSCC), therefore most of the study LICs were selected from DSCC. Additionally, we also chose two LICs from Dhaka North City Corporation (DNCC) to illustrate a comparison with the LICs located in DSCC. Hence, we selected study locations which were located within a distance of 2-3 km from the Eastern trunk main. The study carefully considered distance of the settlements from the proposed trunk main since lack of detailed information regarding user perspectives and way-out to connect, nearby LICs should not be skipped from having sewerage connection”.

  1. Line 370 – the 50-500 BDT range is large, can any further analysis be done here on who exactly was willing to pay 50, and who 500, and in which of the settlements? If limited subsidies or financial support is provided, then those on the lower end of this spectrum will struggle to pay.

Response: Thank you so much for identifying such an important point. We added few lines to elaborate this point under section 3.4  (Line 472-476):

“Among 8 LICs, tenants from Kamalapur Railway Colony, Kamrangirchar and Duaripara wanted to pay within the range of 50 - 100 BDT, while tenants from Tekpara and Dholpur mentioned a range of 200 - 400 BDT, and the tenants from Lalchanmukim Lane, Maniknagar and Nobinbag could afford 300 - 500 BDT as sewerage service fees.”.

  1. As this is a qualitative study (refreshing to see), I would have appreciated a bit more elaboration on why a qualitative approach (and the specific methods) was actually chosen. What is the value of this, compared to quantitative? It’s not just a precursor to quantitative, but should stand in its own right.

Response: Suitability of qualitative research approach in this study added under the section 2.2 (Line 178-182):

“We chose qualitative research approaches so that the study can capture the attitudes of the respondents. One of the core aims of the research was to illustrate user preferences. In this context, qualitative research techniques were seemed best suited, as they are not bound by the methodological limitations and explain something which numbers merely are unable to reveal. These approaches allowed us to be far more speculative about what we have chosen to investigate.”

  1. Did you also account for gender, age, disability, income and other identity markers in the FGDs, aside from house ownership and tenancy? Acknowledgement of these differences, or greater elaboration, would be welcome.

Response: Thank you for asking. We tried to account for gender, age and income but were not successful in all the FGDs. Explanation added under the section 2.4-

We tried to consider gender, age and occupation while choosing our FGD participant groups. However, except tenant groups, in most cases participants were males”.

  1. Did you also distinguish ‘community leaders’ from the house owners or landlords? In my experience, the CBO leaders are often landlords or own multiple houses.

Response: Yes, it’s common that CBO leaders are often landlords or own multiple houses, but we distinguished CBO leaders and conducted FGDs separately (see table 2).

  1. Did you also account for the significant power inequalities in FGDs between landlords, house owners and tenants? I would expect this to have implications for the mixed FGDs.

Response: Thank you for asking. Yes, we account for power inequalities while conducting FGDs with mixed groups.

  1. Table 3 – please clarify terminology both here, and elsewhere in the paper relating to on- site/sewerage/sewers/sanitation/FSM. For example, in Table 3 you mention ‘existing sewerage connections’ – do you mean on-site? It’s not clear.

Response: In this study we didn’t find any area which was connected with a formal sewerage network. The participants generally perceived that sewerage exit pipes those were connected with a storm drain are a sewerage connection. So ‘existing sewerage connection’ means a connection with storm drain based on their perspectives. Moreover, the code also representing current sewerage condition in LICs. We have changed all “existing sewerage connection” to “existing faecal sludge arrangement”

  1. The ordering of some codes was not clear. For example, ‘narrow lanes and connecting points’ were in ‘current sanitation facilities’, but were earlier on referred to under ‘major risk and challenges’ (line 197)

Response: Thank you so much for identifying such error. ‘Narrow lanes and connecting points’ were rightly placed under ‘current sanitation facilities’. But it would be ‘narrow roads’ to mention with ‘major risk and challenges’, as few key personnel identified that narrow roads in some areas are the major obstacles to run cleaning work and at the same time is a major risk to place new sewerage connection under these roads. Here ‘roads’ means connecting roads within an area of the city while ‘lanes’ means paths within a LIC. We have corrected this in the revised version.   

  1. It is not clear what the ticks and crosses mean, and how they relate to the analysis and discussion. Please elaborate or delete them. Greater linkage and referral to Table 3 in the analysis and discussion would be helpful more generally.

Response: Thank you so much for pointing such issue. A single line to clarify that what does the ticks and crosses mean added under the table 3: “Ticks and crosses beside the codes indicating that which codes were frequently mentioned by which groups of the participant”. We also tried to create referral to the table.

  1. Were any other benefits noted beyond ‘health’ and ‘environment’, e.g. convenience, especially for working women and men who have to queue for shared facilities? Privacy issues?

Response: Thank you so much for asking. The study didn’t emphasize on the inconveniences of shared toilet facilities rather it was focused on the consequences for not having a proper sewerage network. We found health benefits and environmental benefits countable within this context. However, thanks for raising this issues. 

  1. Line 135 (reference 22) – what exactly is a ‘transformative worldview’? Why is it ‘transformative’ and relevant? Transformative to / for whom? This relates to the earlier point about whose perspective/s are ultimately central here, the residents (you note in the abstract, quite importantly, that the recommendations come from them) and/or DWASA, who have different goals and perspectives towards ‘slum’ dwellers. With a bit more elaboration, I think the ‘transformative’ worldview would add value to ongoing calls to recognise the urban poor in Dhaka, and across Bangladesh, as entitled urban citizens.

Response: Thank you so much to raise the point. We have deleted ‘transformative worldview’ as this suggestion was made by multiple reviewers.

  1. There are a number of quotes from IG Gate Bank Colony – why so many from this particular LIC? Having a range of quotes from different settlements would nuance the analysis.

Response: Thank you so much for pointing that. We replaced two of such quotes.

“Ours is a hanging toilet. We don’t even have any drain here. So faecal matters stuffed beneath the ground of the toilet and travel with the water we use in the toilet”.

-FGD, Male house owner, Tekpara

 “We are living by the side of a canal. This canal is badly filled with huge wastes. All its need is a 10 minutes of heavy rainfall to overflow of waste in our surrounding. We could have separate drains to pass different wastes in a proper way. If the sewages don’t mix with other wastes and don’t go to the canal, there is a less chance to see them back with the overflow”.

-FGD, Male tenant, Khilgaon Bagicha

  1. Line 334-335 – the DWASA quote underneath the discussion of land tenure insecurity seems a bit disconnected (and focuses more on water). Did DWASA or other KIIs mention anything about land tenure? This seems important.

Response: Thank you so much for reviewing this. That particular quote was uttered by DWASA personnel just to exemplify what DWASA did to provide water connections in the city and to indicate that what they are going to do to provide sewerage connections also. According to the key personnel, land tenure issues handled by City Corporations. We conducted KII with Ward Councilor but we didn’t get any clear decision/plan regarding illegal land tenure in upcoming future.   

  1. Please include a note on ethics – for example, any approvals and whether/how you obtained informed consent from all participants.
    •  

Response: Thank you so much for identifying such a vital issue. We created a new point (2.6) to include a note on ethics.

“Ethical approval

We obtained ethical approval from the Ethical Review Committee (ERC) of the International Centre for Diarrhoeal Disease Research, Bangladesh (icddr,b). Study participants were informed of the aims of the study and their rights. Enumerators read an information sheet to respondents in Bengali, answered any questions raised, and obtained written consent for participation. Respondents were given a copy of the information sheet to keep, and no compensation was provided for participation. Names and numbers were removed from final data sets to protect anonymity.”

  • Minor Points

  1. A brief note on DWASA would be relevant i.e. an autonomous and commercial (for-profit) organization.

Response: Thank you for this suggestion. We added a sentence for DWASA (Line 92-94).

“Dhaka Water Supply and Sewerage Authority (DWASA) is a Bangladesh government agency under the Ministry of Local Government, Rural Development and Co-operatives responsible for water and sewage in Dhaka.”  

  1. The paper is well written but some sections require sharpening to reduce repletion and clarify argument, e.g. lines 165-166, 452-456, and the first three paragraphs of the discussion section.

Response: Thank you so much this important suggestions. We have revised those line s and revised first three paragraph of the discussion.

  1. Line 92-93 (reference 15) – what do you mean by ‘middlemen’? This is a term often used but poorly understood or elaborated (and there is a whole other literature on this). ‘Well functioning informal markets’ is also not clear, what do you mean here? Other citations on pit emptying include: Opel, A. and Bashar, M. K. (2013) Inefficient technology or misperceived demand: The failure of vacutug-based pit- emptying services in Bangladesh, Waterlines. Dhaka, Bangladesh and Zaqout M Cawood S Evans B Barrington B (forthcoming). Sustainable sanitation jobs: prospects for enhancing the livelihoods of pit emptiers in Bangladesh. Third World Quarterly.

Response: Thanks for asking sir. Here ‘middleman’ means the person who negotiate with the pit owners to fix that how much it will cost for emptying. There is a growing market of such emptying service providers, beyond registered sanitation worker and other waste cleaners, which is being operated by these middlemen and termed as ‘Well functioning informal markets’ by the literature. We have included those citations also.   

  1. Line 69-71 (reference 14) – please use original BBS citation

Response: Thanks, we have added original citation.

  1. Table 2 – Please clarify what the ‘existing sewerage connections’ were for tenants?

Response: Thank you for your concern. I am not quite clear about your query. As far I can clarify, among our study areas it was common that exit pipes of the toilets were connected with a storm drain, and a portion of the respondents perceived that connection as sewerage connection. And the respondents whose exit pipes of the toilets were connected with a water body or tank, perceived as toilets without sewerage connection. We have changed all “existing sewerage connection” to “existing faecal sludge arrangement”

  1. Line 93-94 – what % is ‘most’ urban residents?

Response: Thank you for your query. We found that it is 42%; reference added.

  1. Line 94 – why are the treatment plants not operating at full capacity?

Response: Thank you for your interest. We missed it, now added under 1.3.

“However, Dhaka's only treatment plant functions below capacity; due to several blockages and inefficiency issues of the lifting pumps, the treatment plant is not getting enough faecal waste to treat according to its full capacity.”

  1. Line 104 (reference 21) – is this citation on Kibera, Kenya appropriate for the Dhaka focus? Are other references more appropriate?

Response: We have added appropriate references.

  1. Line 174 – what do you mean by ‘status quo’ – surely this varies between and within the LICs studied?

Response: Thank you for your query. We tried to remove confusing words from that sentence.

“These FGDs assessed the satisfaction of the users regarding sewerage facilities in terms of their financial status in different LICs”.

  1. Line 215 – will readers know what vacuum tankers are?

Response: Thank you for identifying. It was wrongly spelled. Now replaced with “Vacutag”

  1. Line 218 – what do you mean by ‘infrastructure components’

Response: Thank you for identifying. Now replaced with “technical issues”

  1. Line 221 – you mention 9 tenant FGDs – were these referring to the ‘sewered’ or ‘non-sewered’ tenants?
    •  

Response: Thank you for your query. It was a mixed response both from the tenant groups having a sewerage pipe connected with storm drain and the tenant groups having sewerage pipe connected with water bodies or tanks.  

  1. Line 236-237 – clarify what is ‘owned’ by the government, the land?

Response: Thanks. We tried to clarify. It was the land which is owned by the government.

  1. Line 362 – only financial situation? Or also occupancy status as a landlord, house owner, tenant etc?

Response: Good point. Thank you. Added accordingly.

  1. Line 442 – 1977 or much earlier? Please clarify if you are referring to the original system or upgrades

Response: Thank you for raising this point. According to the key informant, existing network was built in 1977. We think we are referring to the original system because the current phase of the sewerage master plan is dedicated to upgrade that system.  

  1. Are sources for figures and tables required? Please check journal guidelines

Response: Thank you for raising this point. All figures and tables developed by us.

  1. Line 589 – remove ‘Environment & Urbanization: World leading environmental and urban studies journal. p.10’ from reference (12) (incorrect).

Response: Thanks for finding. Suggested part is removed.

Reviewer 2 Report

I enjoyed reading the manuscript. Overall, this is a well-written manuscript. The introduction is relevant that provides sufficient information, but revisions are required for readers clarity. The methods, especially data analysis approaches are generally appropriate. I recommend that additional clarification for the transformative worldview as an analysis tool should be provided. Findings are comprehensive, but some of them are reported without citing the accurate number of respondents or FGDs. Thus, findings need to be well linked to the data as a major revision. The discussions can be improved by complementing them with some additional feedback from similar previous scholarly studies. Separate comments have been included in the manuscript. 

Author Response

Response to reviewer # 2 comments

I enjoyed reading the manuscript. Overall, this is a well-written manuscript. The introduction is relevant that provides sufficient information, but revisions are required for readers clarity. The methods, especially data analysis approaches are generally appropriate. I recommend that additional clarification for the transformative worldview as an analysis tool should be provided. Findings are comprehensive, but some of them are reported without citing the accurate number of respondents or FGDs. Thus, findings need to be well linked to the data as a major revision. The discussions can be improved by complementing them with some additional feedback from similar previous scholarly studies. Separate comments have been included in the manuscript. 

  1. Line 20-22: These two sentences are not linked. The authors could present country scale data in the first sentence.

Response: Thanks for asking. Revised.

“In Bangladesh, approximately 31% of urban residents are living without safely-managed sanitation, the majority of whom are slum residents”.

  1. Line 35: Please check whether all keywords are used in the abstract.

Response: Thanks for this suggestion. We have checked and found that all key words were used in abstract except affordability and feasibility. These two words were not used in abstract but the similar theme or idea were described in the abstract. However, we have dropped the word “feasibility” from the keywords.

  1. Line 43: Need to define - safely managed, sanitation services, basic services and limited services.

Response: Thanks for asking. Definition added.

“According to the WHO-unicef Joint Monitoring Program (JMP), improved sanitation facilities are those designed to hygienically separate excreta from human contact; improved sanitation facilities which are not shared with other households and the excreta produced are treated properly considered safely managed sanitation services. If the excreta from improved sanitation facilities are not safely managed then those facilities are classified as a basic sanitation service; people using improved facilities which are shared with other households is classified as a limited service.”

  1. Line 48: Need to define low and middle income for the study context.

Response: Thanks for identifying. Definition added in Line 56-57:

“The Bangladeshi low and middle income communities usually earn between $2 to $20 per capita per day

  1. Line 53 -55: Need a transition sentence here to get readers prepared to follow the following sentence.

Response: Thanks for identifying. A sentence to make transition is added.

“Generally these costs are funded both from the government along with other international agencies such as different development banks”.  

  1. Line 55: Authors may want to change the word here.

Response: Thanks. The word “we” removed.

  1. Line 115: When was the study conducted?

Response: Thank you for asking. Duration of the study added (Line 157-158).

The study was conducted from February 2019 to February 2020”.

  1. Figure 1: Source?

Response: Thank you for asking. This figure was illustrated by us.

  1. Line 135: Do FGDs participants selection process ensure that they represent undervalued groups?

Response: Thank you for asking. We think that FGDs participants’ selection process ensures that they represent all groups from the LIC including undervalued groups.  We added in Line 234-235:

We tried to consider gender, age and occupation while choosing our FGD participant groups”

  1. Line 151: What formats were used? i.e open ended, semi-structured etc?

Response: Thank you for asking. We completed the sentence after addressing your comment (Line 208-209)-

“Following an open-ended questionnaire we conducted KIIs with the officials who were directly involved with DSIP and were actively engaged with the existing and proposed sewerage treatment plant”.

  1. Line 170: The authors could discuss the process followed to conduct FDGs.

Response: Thank you for this suggestion. We have elaborated the section 2.4 and described process of conducting FGDs.

  1. Line 229: Interview data and place?

Response: Thank you for identifying. Place was mentioned, interview data added.

  1. Line 243: Interview data and place?

Response: Thank you for identifying. Place was mentioned, interview data added.

  1. Line 251: Interview data and place?

Response: Thank you for identifying. Place was mentioned, interview data added.

  1. Line 261: Which groups of tenants and how many said this?

Response: Thank you for noticing. Tried to re-write (Line 348, 349):

“Generally, in all study areas, all the tenants of each groups”

  1. Line 270: How many of them perceived so?

Response: Thank you for noticing. Tried to mention the number (Line 358:

“In 5 FGDs”.

  1. Line 272: Why they perceived so?

Response: Thank you for noticing. Tried to add their perception (Line 361);

“if a proper sewerage connection can be installed.”

  1. Line 278: Interview data and place?

Response: Thank you for identifying. Place was mentioned, interview data added

  1. Line 287: How many of these interviewed hypothesised?

Response: Thank you for identifying. Numbers of the FGDs are added (Line 382):

“Along with these health consequences, respondents in 5 FGDs with tenants and 6 FGDs with landlord/homeowners”

  1. Line 293: How many accurately?

Response: Thank you for identifying. Numbers of the FGDs are added (Line 389):

“In 6 FGDs out of 10 FGDs with the tenant groups, respondents”

  1. Line 323: How and how many?

Response: Thank you for noticing. Respondent type and number added (Line 419):

“Landlord/homeowner groups in 3 FGDs and tenants in 2 FGDs.”

  1. Line 341: Interview data and place?

Response: Thank you for identifying. Place was mentioned, interview data added.

  1. Line 372: How many of them?

Response: Thank you for noticing. Number added Line 477: 8 tenants groups). Names also mentioned in previous sentences.

  1. Line 380: Who is this group?

Response: Thank you for asking. We added the group name in Line 485:

“Two tenant’s groups in Lalchanmukim Lane and Shyampur were found buying.”

  1. Line 407: Interview data and place?

Response: Thank you for identifying. Place was mentioned, interview data added.

  1. Line 418-420: Were these reported uniformly reported in all 9 KIIs?

Response: Thank you for asking. Different barriers were mentioned differently in all the 9 KIIs but these barriers were common and were frequently mentioned as major risks and challenges. That’s why we wrote “based on 9 KIIs”.  

  1. Line 431: Interview data and place?

Response: Thank you for identifying. Place was mentioned, interview data added.

  1. Line 450: found in the study?

Response: Thank you for your query. This sentence was written based on the information provided by key informants of DWASA who are actively engaged with the DSIP project.  

  1. Line 478: They are the sub-groups of respondents. What others recommended?

Response: Thank you for your query. We concluded these recommendations here only from that sub groups. Other major recommendations from the rest of the groups are placed in the later part.

  1. Line 485-487: How does the previous study respond to these findings?

Response: Thank you for your query. We tried to add a previous study which talked about affordability range to pay bills for wastewater.

“A community’s average cost of water and wastewater services within a municipality’s district is measured as a percentage of the median household income within the city limits. If this value is greater than 4.0% or 4.5% for both water and wastewater services, the system is considered as a high cost and not affordable for families. Consistent with these affordability parameters, our study findings confirm four different preferences which were commonly identified as a possible payment method.”

  1. Line 506: What percentage of participants?

Response: Thank you for identifying. We added the number of FGDs, Line 632 (8 LICs).

  1. Line 525: Both FGDs and informants?

Response: Thank you for identifying. We talked about FGD participants only. Added in the manuscript.

Reviewer 3 Report

The paper presents a study on the perceptions of community members and sanitation stakeholders on opportunities and challenges associated with connecting low-income communities to a new sewerage network being planned/built in Dhaka, Bangladesh.

Connecting low-income communities to sewerage network is a major step needed to achieve citywide inclusive sanitation. It is also a very challenging one, so research in this area is very welcome.

The findings in this paper are worth publishing. However major revisions are needed, particularly for readability. My overall comments:

  1. The authors' use of the term "sewerage network" and other related terms is confusing. I can't tell if the participants currently have a real sewerage network or if they have on-site sanitation facilities that discharge to drains and sometimes on-site containment. I suspect it's the latter and the authors are erroneously calling the discharge of toilets to drains a "sewerage network", but it's really hard to tell. It will be much easier for readers if the terms "sewerage network" and "sewers" are only used to refer to the proposed sewerage network as part of the DSIP. If faecal sludge is actually managed via pits, tanks, and direct discharge to storm drains, please refer to this generally as the "current faecal sludge management arrangement".
  2. The discussion section requires major revisions. Currently, it mostly just repeats information from the results section. Most of the discussion section can be cut out entirely, and should be replaced with the author's thoughts on the implications of what they found for policy, practice or research.

 My specific comments are below:

Line 6: I’m not sure any of the listed participants are actually “policymakers”. Perhaps this should be “stakeholders” instead.

Line 23: Per my comment below about lines 101-102, it seems like this study is more about the perceptions of challenges and opportunities of connecting to a sewerage network, and not an assessment of feasibility per se.

Lines 41-42: Please provide a reference for global breakdown of levels of urban sanitation service.

Lines 47-48: It doesn’t seem accurate to say that it is common in high-income areas globally (meaning including in Europe, North America, etc.) for flush-toilets to be discharged directly into drains with no containment (I also could not find this claim in the reference). Perhaps re-phrase this to specify low- and middle-income countries.

Line 61: Because “low-income communities” is the central unit of analysis in this paper, I think it is important to define what the authors mean by this somewhere. In this line it stated that 35% of Dhaka’s urban population lives in low-income communities, so perhaps it could be defined here (or if not here, the authors should define it somewhere).

Line 91: Do you mean sanitation or FSM services here instead of sewerage services? Sewerage services pertain specifically to sewers – sewers do not exist outside of Dhaka and are not operated by informal markets.

Lines 101-102: It seems like this study assesses perceptions on opportunities and challenges of connecting LICs to a sewer network instead of an assessment of feasibility. It sounds like the authors are suggesting they did a feasibility study (i.e. assessed the practicality of a project) which doesn’t seem to be the case.

Line 114: Somewhere in this section, please include a statement on what ethical clearance/approval was acquired to conduct this research.

Lines 116-120: I am a bit confused, have the two trunk mains already been built, or are they still being planned? Please clarify.

Lines 134-135: “Data was conceptualised and narrated following the transformative worldview” – I don’t understand what this sentence means. I have feeling it will be trouble than it’s worth to try to clarify so I suggest removing it (or alternatively, explaining in more detail what this means).

Line 142: Define CBO as community-based organisation here (first time CBO is used).

Line 172-173: I’m confused about the authors’ use of the term “sewer”. A sewer is commonly used to refer to a pipe or conduit that is part of a sewerage network (i.e. as being implemented by DWASA). By “sewer connection to the storm drainage” do you mean a toilet that flushes directly to an open drain (i.e. no containment)? What does “tenants without a sewer connection” mean? I thought no one in the study communities is connected to a sewer yet (hence this purpose of this whole study)?

Figure 2: What does the box “validating the accuracy of the information” mean here? The authors do not explain in the text. In qualitative research, validation typically means going back to the participants and checking whether the analysis accurately represents what they meant to say, but the authors don’t mention doing this.

Table 3: I appreciate the transparency in showing the processed data, but I wonder if this table is a bit much for the paper itself. I wonder if it should be included as supplementary information instead of being included in the paper.

Line 206: What does “reach the highest score” mean? No scoring system has been mentioned before. Does it mean these codes were used the most often?

Line 213: Again, I am confused what the authors mean by “current sewerage system”. Do they mean faecal sludge management system?

Line 216: Can the authors clarify if the storm drains are covered or uncovered?

Lines 218-220: What are these pipes that are being blocked or leaking? Are these pipes connecting toilets to pits and/or septic tanks?

Line 219: I’m not sure I understand what is mean when a term or concept “overlaps” in the data. Overlaps with what? If ‘pipe blockage’ overlaps 51 times, does that mean participants mentioned it 51 times over the course of data collection?

Lines 223-226: Was the overflow of faecal waste coming from the existing sewerage network or from on-site pits and tanks? In section 1.2, it was stated that faecal matter spilling from pits and tanks was known to be a major issue.

Line 299: After reading this section, it seems like the section is more about “willingness to connect to a sewerage network” than feasibility (although the last paragraph touches on technical feasibility).

Line 311: I think these should be called five key “conditions” instead of five key “criteria”.

Lines 318-319: I am having a hard time distinguishing between “household size/type based service charge”, “area-based subsidies” and “income-based subsidies” based on the descriptions in this paragraph. Do the authors mean to say that participants suggested discounting service charges for users based on:

  1. The size of the household or type of toilet used;
  2. The income of the household; or
  3. The geographic area where the household is located?

Lines 343-344: This line about legalising illegal residences is a different point than the rest of the paragraph (which is about technological solutions) – I’m not sure why it is mentioned here, it doesn’t sound like the participants mentioned it. Perhaps remove this line, and discuss legalisation issues elsewhere in the paper.

Lines 360-391: This is a very long paragraph that groups together discussion on various forms of payment mechanisms for various expenses (service connections, ongoing service charges, water, maintenance). I suggest breaking this paragraph into multiple paragraphs and clearly signposting which expenses are being discussed. Same with the following paragraph.

Lines 440-441: I assume you mean ML (megalitre). ml means millilitre!

Line 448: The discussion section requires major revisions. The purpose of a discussion section is for the authors to discuss what they think are the implications of the results for practice, policy, research, etc. In its current form, this section mostly repeats the same information in the results section with a smattering of new results dispersed throughout.

Author Response

Response to reviewer # 3 comments

The paper presents a study on the perceptions of community members and sanitation stakeholders on opportunities and challenges associated with connecting low-income communities to a new sewerage network being planned/built in Dhaka, Bangladesh.

Connecting low-income communities to sewerage network is a major step needed to achieve citywide inclusive sanitation. It is also a very challenging one, so research in this area is very welcome.

The findings in this paper are worth publishing. However major revisions are needed, particularly for readability. My overall comments:

  1. The authors' use of the term "sewerage network" and other related terms is confusing. I can't tell if the participants currently have a real sewerage network or if they have on-site sanitation facilities that discharge to drains and sometimes on-site containment. I suspect it's the latter and the authors are erroneously calling the discharge of toilets to drains a "sewerage network", but it's really hard to tell. It will be much easier for readers if the terms "sewerage network" and "sewers" are only used to refer to the proposed sewerage network as part of the DSIP. If faecal sludge is actually managed via pits, tanks, and direct discharge to storm drains, please refer to this generally as the "current faecal sludge management arrangement".

Response: Thank you so much for your effective suggestion. We made several replacements according to your suggestion. We have changed all “existing sewerage network” to “existing faecal sludge arrangement to storm drainage”.

  1. The discussion section requires major revisions. Currently, it mostly just repeats information from the results section. Most of the discussion section can be cut out entirely, and should be replaced with the author's thoughts on the implications of what they found for policy, practice or research.

  • Response: Thank you so much, we have revised the entire discussion section.

Specific comments:

  1. Line 6: I’m not sure any of the listed participants are actually “policymakers”. Perhaps this should be “stakeholders” instead.

Response: Thank you so much for your effective suggestion. We replaced the word as per your suggestion.

  1. Line 23: Per my comment below about lines 101-102, it seems like this study is more about the perceptions of challenges and opportunities of connecting to a sewerage network, and not an assessment of feasibility per se.

Response: Thank you so much for your effective suggestion. We made several replacements according to your suggestion (Line 142, 143).

  1. Lines 41-42: Please provide a reference for global breakdown of levels of urban sanitation service.

Response: Thank you so much for identifying. Reference added.

  1. Lines 47-48: It doesn’t seem accurate to say that it is common in high-income areas globally (meaning including in Europe, North America, etc.) for flush-toilets to be discharged directly into drains with no containment (I also could not find this claim in the reference). Perhaps re-phrase this to specify low- and middle-income countries.

Response: Thank you so much for identifying such an issue. We tried to re-phrase the sentence.

  1. Line 61: Because “low-income communities” is the central unit of analysis in this paper, I think it is important to define what the authors mean by this somewhere. In this line it stated that 35% of Dhaka’s urban population lives in low-income communities, so perhaps it could be defined here (or if not here, the authors should define it somewhere).

Response: Thank you so much for identifying such an issue. We added an income range to mark up low income communities in section 1.1.

  1. Line 91: Do you mean sanitation or FSM services here instead of sewerage services? Sewerage services pertain specifically to sewers – sewers do not exist outside of Dhaka and are not operated by informal markets.

Response: Thank you so much for identifying such an issue. We revised as FSM.

  1. Lines 101-102: It seems like this study assesses perceptions on opportunities and challenges of connecting LICs to a sewer network instead of an assessment of feasibility. It sounds like the authors are suggesting they did a feasibility study (i.e. assessed the practicality of a project) which doesn’t seem to be the case.

Response: Thank you so much for identifying such an issue. We dropped the word feasibility.

  1. Line 114: Somewhere in this section, please include a statement on what ethical clearance/approval was acquired to conduct this research.

Response: Thank you so much for your suggestion. Section 2.6 added.

  1. Lines 116-120: I am a bit confused, have the two trunk mains already been built, or are they still being planned? Please clarify.

Response: Thank you for your interest. The two trunk main was still in planning position during the study; they are not completely built yet.

  1. Lines 134-135: “Data was conceptualised and narrated following the transformative worldview” – I don’t understand what this sentence means. I have feeling it will be trouble than it’s worth to try to clarify so I suggest removing it (or alternatively, explaining in more detail what this means).

Response: Thank you for your potential suggestion. We have dropped that as multiple reviewers raise the issues.

  1. Line 142: Define CBO as community-based organisation here (first time CBO is used).

Response: Thank you for bringing such mistake in front. Definition added.

  1. Line 172-173: I’m confused about the authors’ use of the term “sewer”. A sewer is commonly used to refer to a pipe or conduit that is part of a sewerage network (i.e. as being implemented by DWASA). By “sewer connection to the storm drainage” do you mean a toilet that flushes directly to an open drain (i.e. no containment)? What does “tenants without a sewer connection” mean? I thought no one in the study communities is connected to a sewer yet (hence this purpose of this whole study)?

Response: We strongly agree with you on that point that no one in the study communities is connected to a sewer yet. And you are right that “sewer connection to the storm drainage” means a toilet that flushes directly to a storm drain. On the other hand, “tenants without a sewer connection” means a toilet that flushes to an open water body anyway. We have changed all “existing sewerage connection” to “current faecal sludge arrangement to storm drainage”

  1. Figure 2: What does the box “validating the accuracy of the information” mean here? The authors do not explain in the text. In qualitative research, validation typically means going back to the participants and checking whether the analysis accurately represents what they meant to say, but the authors don’t mention doing this.

Response: Thank you for raising that issue. We added regarding how we validate the data accuracy -

“We also cross checked our data during fieldwork to ensure its validity while talking to different respondents in different areas. During organising, coding, interpretation, and in all stages of data analysis, we did not intend to misinterpret any data prioritising our viewpoints”

  1. Table 3: I appreciate the transparency in showing the processed data, but I wonder if this table is a bit much for the paper itself. I wonder if it should be included as supplementary information instead of being included in the paper.

Response: We have moved that table as a supplementary material.

  1. Line 206: What does “reach the highest score” mean? No scoring system has been mentioned before. Does it mean these codes were used the most often?

Response: Thank you. You guessed rightly. We meant that these codes were used most often by “reach the highest score”. We have replaced that as codes used most often.

  1. Line 213: Again, I am confused what the authors mean by “current sewerage system”. Do they mean faecal sludge management system?

Response: We have changed all “existing sewerage connection” to “current faecal sludge arrangement to storm drainage”

  1. Line 216: Can the authors clarify if the storm drains are covered or uncovered?

Response: Thank you for asking. We tried to clarify this under section 3.1 -

“Among 16 LICs, we found storm drains completely open in 3 LICs, yet several toilet exit pipes were connected with those drains. In 11 LICs storm drains were mostly covered but still having minor uncovered sections. We didn’t found any storm drainage in the rest LICs”.

  1. Lines 218-220: What are these pipes that are being blocked or leaking? Are these pipes connecting toilets to pits and/or septic tanks?

Response: Thank you for asking. These are the connecting pipes. In most cases, according to the respondents, these pipes are connecting toilets to the drain or the place where fecal matters finally goes.

  1. Line 219: I’m not sure I understand what is mean when a term or concept “overlaps” in the data. Overlaps with what? If ‘pipe blockage’ overlaps 51 times, does that mean participants mentioned it 51 times over the course of data collection?

Response: Thank you for understanding from our perspectives. How many times a code overlaps means the participants mentioned it how many times over the course of data collection.  We have replaced all “overlaps” by “mentioned”.

  1. Lines 223-226: Was the overflow of faecal waste coming from the existing sewerage network or from on-site pits and tanks? In section 1.2, it was stated that faecal matter spilling from pits and tanks was known to be a major issue.

Response: Thank you for raising this issue. To gain readability we re-phrased the sentence -

“Nine tenant FGDs had mentioned that during rainy season, there was frequent overflow inside their household area from the drains and canals where faecal matters finally exits.”

  1. Line 299: After reading this section, it seems like the section is more about “willingness to connect to a sewerage network” than feasibility (although the last paragraph touches on technical feasibility).

Response: we have changed the section title as willingness to connect to a sewerage network

  1. Line 311: I think these should be called five key “conditions” instead of five key “criteria”.

Response: Thank you for your suggestion. We replaced “criteria” with “conditions”.

  1. Lines 318-319: I am having a hard time distinguishing between “household size/type based service charge”, “area-based subsidies” and “income-based subsidies” based on the descriptions in this paragraph. Do the authors mean to say that participants suggested discounting service charges for users based on:
  1. The size of the household or type of toilet used;
  2. The income of the household; or
  3. The geographic area where the household is located?

Response: Thank you for understand rightly. Through different FGDs the participants suggested these subsidies. They suggested discounted service charges based on their household type, financial status and the areas where they are living in poor condition.

  1. Lines 343-344: This line about legalising illegal residences is a different point than the rest of the paragraph (which is about technological solutions) – I’m not sure why it is mentioned here, it doesn’t sound like the participants mentioned it. Perhaps remove this line, and discuss legalisation issues elsewhere in the paper.

Response: Thank you for identifying that. We understand your point. The fact is that most of the ‘possible strategies’ mentioned in KIIs can be affected if the legalization issues remain unsolved. Based on this relevancy we decided to put this line here. But this sentence is deleted now.

  1. Lines 360-391: This is a very long paragraph that groups together discussion on various forms of payment mechanisms for various expenses (service connections, ongoing service charges, water, maintenance). I suggest breaking this paragraph into multiple paragraphs and clearly signposting which expenses are being discussed. Same with the following paragraph.

Response: Thank you for your incredible suggestion. We split the paragraph into 3. We were aiming to focus on the user’s affordability here. Data regarding other expenses were collected to provide an understanding that how much they already spending now, which can indicate how much they can afford in future for such utilities. 

  1. Lines 440-441: I assume you mean ML (megalitre). ml means millilitre!

Response: Thank you for highlighting such a big error. It will be “ML”, replaced.

  1. Line 448: The discussion section requires major revisions. The purpose of a discussion section is for the authors to discuss what they think are the implications of the results for practice, policy, research, etc. In its current form, this section mostly repeats the same information in the results section with a smattering of new results dispersed throughout.

Response: We have revised the entire discussion section.

Round 2

Reviewer 3 Report

Thanks to the authors for thoroughly responding to my comments. I am satisfied with the changes. I have just one final suggestion - I think the following line can be removed:

"During organising, coding, interpretation, and in all stages of data analysis, we did not intend to misinterpret any data prioritising our viewpoints"

I hope it goes without saying that you did not intentionally misrepresent data!